# *Ganoderma lucidum* Effects on Mood and Health-Related Quality of Life in Women with Fibromyalgia

**DOI:** 10.3390/healthcare8040520

**Published:** 2020-11-30

**Authors:** Francesco Pazzi, José Carmelo Adsuar, Francisco Javier Domínguez-Muñoz, Miguel Angel García-Gordillo, Narcis Gusi, Daniel Collado-Mateo

**Affiliations:** 1Health Economy Motricity and Education (HEME), Faculty of Sport Science, University of Extremadura, 10003 Cáceres, Spain; frapaz76@gmail.com; 2Physical Activity and Quality of Life Research Group (AFYCAV), Faculty of Sport Science, University of Extremadura, 10003 Cáceres, Spain; fjdominguez@unex.es (F.J.D.-M.); ngusi@unex.es (N.G.); 3Facultad de Administración y Negocios, Universidad Autónoma de Chile, 3467987 Sede Talca, Chile; miguel.garcia@uautonoma.cl; 4Centre for Sport Studies, Rey Juan Carlos University, 28943 Madrid, Spain; danicolladom@gmail.com

**Keywords:** complementary and alternative medicine, happiness, depression, satisfaction with life, global impression of improvement, experimental study

## Abstract

Fibromyalgia syndrome is a chronic rheumatic disorder characterized by generalized and widespread musculoskeletal pain. It is associated with several secondary symptoms such as psychological and pain-specific distress, which can directly impact daily functioning and quality of life, like anxiety and depression. The *Ganoderma lucidum* (GL) mushroom seems to be able to improve fibromyalgia symptoms, including depression and pain. The purpose of the study is to evaluate the effects of GL on happiness, depression, satisfaction with life, and health-related quality of life in women with fibromyalgia. A double-blind, randomized placebo pilot trial was carried out, with one group taking 6 g/day of micro-milled GL carpophores for 6 weeks, during which the second group took a placebo. Our results did not show any statistically significant between-group differences, although a distinct trend of improved levels of happiness and satisfaction with life and reduced depression were evident at the end of treatment compared to the baseline in the GL group. However, due to the limitations of the study protocol, additional studies are necessary to confirm these findings.

## 1. Introduction

Fibromyalgia syndrome (FMS) is a chronic rheumatic disorder of which its etiology is not fully understood; it is characterized by generalized and widespread musculoskeletal pain. It is associated with several secondary symptoms such as fatigue, impaired sleep, and other psychological and pain-specific distress, which can directly impact daily functioning and quality of life, like anxiety and depression [1]. Its prevalence in European populations oscillates from 0.5% to 5.0% [2], and it imposes significant healthcare and societal burdens [3].

A variety of pharmacological and nonpharmacological treatments have been proposed to treat FMS due to the absence of any definitive strategy [4]. The most widely accepted approach consists of a multidisciplinary approach based on physical exercise, education, and behavioral techniques, combined with several drug treatments, including antidepressives and muscular relaxants [1]. These pharmacological treatments are likely to have a wide list of potential side effects. In view of this, complementary and alternative medicine could be helpful to treat patients with FMS [5].

*Ganoderma lucidum* (GL), commonly known as Lingzhi or Reishi, is a fungus with more than 2000 years of history of safe use for its health properties. These properties seem to indicate that GL could alleviate some of the principal symptoms of FMS. In particular, it has been reported that GL could have analgesic and sedative effects, may improve the quality and duration of sleep, and could reduce fatigue and depression [6] and improve physical fitness in women with fibromyalgia [7]. Finally, previous research has shown that taking GL as nutritional supplementation is cost-effective in Spanish women with FMS [8].

All these findings suggest that GL could have a positive impact on mood and health-related quality of life (HRQoL). Furthermore, given the antioxidant properties of GL, its use could be expected to have a positive effect on the impact of FMS since oxidative stress may be implicated in the pathophysiology of the syndrome [9].

The purpose of this paper is to assess the effects of GL on happiness, depression, the Satisfaction with Life Questionnaire (SWL), HRQoL, and the perception of change in women with FMS. To our knowledge, to date, no studies have been carried out to assess such variables in patients with FMS.

## 2. Materials and Methods

### 2.1. Participants

Participants were recruited from three FMS associations in Spain. Enrolment began in October 2014, and data collection was completed in January 2015. Patients were eligible for inclusion if they fulfilled the following criteria: (a) aged 18 or over; (b) diagnosed with FMS by a rheumatologist and meeting the American College of Rheumatology 1990 criteria for FMS; (c) able to communicate effectively with the study staff; (d) had given written informed consent. Patients were excluded if they fulfilled the following criteria: (a) pregnant; (b) changed their usual care therapy regime during the 6 weeks of the trial treatment; (c) taking immunosuppressants; (d) suffering from diabetes; (e) participating in other studies; (f) taking vitamin C supplements or anticoagulants; (g) had taken GL before the study started.

Details of participants and reasons for drop out are shown in the flow diagram in Figure 1 a total of 70 subjects (67 women and 3 men) responded and requested additional information. Of these, six were subsequently excluded: one declined to take part in the study, one had not been diagnosed with FMS, one did not meet the American College of Rheumatology 1990 criteria for FMS, and three suffered from diabetes. Finally, a total of 64 women with FMS participated in the study and provided written informed consent in accordance with the Declaration of Helsinki. All patients continued with their usual treatment during the study. The Committee of Bioethics of the University of Extremadura (Spain) approved the study, and it was registered in the Australian New Zealand Clinical Trials Registry (ANZCTR), ID: ACTRN12614001201662.

A double-blind, randomized placebo pilot trial was carried out. Participants were randomly assigned to either the *Ganoderma lucidum* group (GLG; *n* = 32) or the placebo group (PG; *n* = 32). Randomization was performed by a research assistant using a random number table from which each participant was given a code number. This researcher did not take part in the acquisition or statistical analysis of data. Neither the participants nor the investigators were aware of the group allocation.

### 2.2. Instruments

Data collection, before and after treatment, was performed at the headquarters of each local association.

Happiness was assessed using the Subjective Happiness Scale (SHS). This is a 4-item instrument rated on a 1–7 Likert-type scale that measures global subjective happiness by means of statements which participants use to either self-rate themselves or compare themselves to others. The first two items require the individual to describe themselves in general and compare themselves with their peers, while the other two items present brief descriptions of happy or unhappy individuals and the interviewees are asked to indicate the degree to which they identify with the descriptions. This instrument has been translated into Spanish and validated with a Spanish adult cohort [10].

SWL was assessed using the Satisfaction with Life Scale (SWLS). This is a 5-item scale with ratings ranging from 5 to 25, with higher scores reflecting greater cognitive well-being. It is a valid and reliable measure of life satisfaction within the Spanish context [11], and it is commonly used in patients with fibromyalgia [12].

Depression symptoms were evaluated through the Geriatric Depression Scale (GDS), which has been validated in the Spanish population [13] and seems to be the most appropriate depression questionnaire in FMS research because it is less focused on somatic symptoms than other depression questionnaires [14].

General HRQoL was measured using the Medical Outcomes Study Short Form-12 questionnaire (SF-12), version 2. It is an abbreviated version of the SF-36 that consists of 12 items covering the physical and mental aspects of health. It comprises 6 dimensions: *physical functioning, role limitations, social functioning, pain, mental health, and vitality*. The SF-12 questionnaire has been demonstrated to be a practical alternative to the SF-36 for the Spanish population [15]; its reliability and validity has also been demonstrated [16].

Patients′ perception of change was assessed following treatment (both GLG and PG) using the Global Impression of Improvement Scale (GIIS) [17], where scoring ranges from 7 (*very much worse*) to 1 (*very much improved*). This scale has been shown to be highly correlated with change in pain variables in patients with FMS [18].

Participant’s adherence to the treatment was monitored through a weekly telephone call, where patients were asked how many doses they had taken that week and if they had experienced any problems: all participants had been provided with a notebook to record the doses taken and any issues that arose. As another check, participants were asked to return all vials at a postmeasures meeting so that the number of doses taken and those missed could be verified.

In this way, weekly checks were made on safety and toxicity, as well as adherence.

### 2.3. Procedure

GLG received 6 g/day (divided into 2 equal doses) of micromilled carpophores of GL for 6 weeks. Based on the literature consulted [19,20], this was the minimum effective dose and it was chosen to avoid possible adverse effects that could result from an overdose, taking into consideration the chemical hypersensitivity that can affect women with fibromyalgia [21]. Both GL and the placebo used were provided by the company “MundoReishi Salud S.L.” (Palencia, Spain) in vials containing 3 g of either GL or placebo. Patients were asked to dissolve the GL or placebo in warm water and to ingest it orally just before breakfast and dinner.

The placebo was composed entirely of *Ceratonia siliqua* (CS) flour, which was chosen for its similarity in color and texture to GL and for the absence of possible effects on the outcome measures.

The research assistant who distributed the doses did not participate in the acquisition or statistical analysis of data. Participants were telephoned once a week to check their progress and to spontaneously resolve any potential doubts.

### 2.4. Sample and Statistical Analysis

All statistical analyses were performed using IBM SPSS v21 (IBM, Armonk, NY, USA). Means and standard deviations of descriptive variables were calculated in order to characterize the two groups. Student′s *t*-test for independent samples and the chi-squared test for categorical variables were used to compare the characteristics of GLG and PG at baseline. The distribution of data was checked using the Kolmogorov–Smirnov test with Lilliefors’ significance.

An analysis of variance (ANOVA) for repeated measures was used to calculate the effects of the treatment on the outcomes of the study: happiness, SWL, depression, and HRQoL. To reduce the probability of making type I errors, since multiple hypotheses were being tested, the statistical significance of *p* was calculated with the correction of Bonferroni, this being equal to 0.05/12 = 0.004. A paired *t*-test was employed to estimate the changes for the two groups as compared to the baseline. In addition, a Student’s *t*-test was used to compare GIIS scores between GLG and PG.

Two different analyses were performed. The first comprised participants who fulfilled all inclusion criteria and completed the study, taking at least 80% of the doses (*n* = 50). The second analysis was the intent-to-treat analysis; it comprised the 64 initial participants and utilized the data of all participants that came to the post-treatment measures, including data from those who took less than 80% of the dose (*n* = 10). The post-treatment data of the remainder of the sample (*n* = 4) were imputed according to the mean change of their group. Given that some participants did not answer SHS (*n* = 1), SWLS (*n* = 1), or GDS (*n* = 3), their score in this specific questionnaire was also imputed. The level of significance was set at *p* < 0.05.

The face-to-face interviews to complete the various questionnaires lasted approximately 45 min per patient (range: 30 to 75 min).

## 3. Results

Statistically significant differences between the two groups at baseline were observed only in terms of educational qualifications (Table 1).

A total of 52 participants took at least 80% of the treatment (Figure 1). However, two participants were later excluded because they started receiving other nonstandard care therapies. A total of 50 participants answered the question about the perception of treatment efficacy. In addition, one participant was not able to complete SHS and SWLS because of emotional impairment (*n* = 49). Similarly, three participants were not able to complete GDS for the same reason (*n* = 47).

Following the 6-week treatment period, after applying the Bonferroni correction (*p* < 0.004), no statistically significant difference was found between GLG and PG in any of the outcome measures (Table 2 and Table 3). However, a paired *t*-test revealed a distinct trend in terms of improvement between the start and end of treatment in GLG with respect to SHS, depression, and SWL scores, whereas no such tendency was found in PG.

The number-needed-to-treat analysis for GDS was four. This means that about one in four patients benefited from the treatment. In terms of SWLS, this number was three, meaning that the treatment resulted in improvement for about one in three patients.

Out of the 64 participants, 52 took at least 80% of the treatment, representing 81% of the initial sample. The number of participants who decided to stop treatment was 5 for both groups. The total abandonment percentage due to treatment was less than 16%, and no serious adverse effects were experienced by any of these participants, only mild discomfort (Table 4). The two most common issues were stomach problems and nausea.

## 4. Discussion

To our knowledge, this is the first study assessing the effects of GL on the happiness, depression level, SWL, and HRQoL of FMS patients. Previous findings in different populations have suggested that the properties of GL could alleviate some of the principal symptoms of FMS and improve HRQoL. In contrast to these findings, we did not find any statistically significant difference between GLG and PG for any of the outcome measures. However, after the 6-week treatment, we did notice an improvement in the happiness, SWL, and GDS scores.

The treatment effect on general happiness was 17.2%, whereas the improvement relative to the baseline was close to 22%. However, why should GL improve happiness in FMS patients? This question might be answered by considering the results in the different dimensions of the SF-12 questionnaire. Although no statistically significant difference was found in the between-group analysis for any dimension of the SF-12 questionnaire, the paired *t*-test analysis revealed that GLG experienced an improvement in bodily pain, general health, social functioning, emotional role, and mental health. All these improvements could have led to an enhancement of happiness levels. In particular, for the bodily pain dimension assessed by SF-12, pain level changes relative to baseline were 40%, whereas the treatment effect compared to the placebo was 25%. These results are far from the improvement of around 21% reported as the placebo effect in previous studies [22].

Depression was enhanced in the intragroup analysis in GLG, but no statistically significant differences were observed in the between-group analysis. These results are in contrast with those found by Zhao et al. [23], where GL spore powder was used in patients with breast cancer. The difference with our results could be explained by the fact that spores can concentrate more active compounds on depression than the wall fruit body utilized in our study. In fact, the administered dose was lower than what we used in this trial: 3 g/day compared to 6 g/day.

Similar results to those observed in depression were found for SWLS. No statistically significant differences were observed in the between-group analysis, while a significant improvement was found in the between-group analysis. The SWLS score at baseline was slightly higher than the score reported in previous studies with Spanish FMS patients [12]. Those results may indicate an important effect of GL on the mood of FMS patients, given that the placebo did not affect any of the assessed mood variables.

GLG perceived that the treatment had an efficacy of 2.54 in GIIS score, which means the improvement fell between a score of 2 (much improved) and 3 (minimally improved). On the other hand, PG perceived an efficacy score of 3.46, where 4 indicates “no change” and 3 “minimally better” [17]. These findings are in accordance with those of Tang et al. [24], where the patient’s perception of change was measured through the Clinical Global Impression Scale.

The reason behind this perception was studied: a statistically significant correlation between GIIS score and change in depression levels in GLG was found (*R* = 0.52; *p* < 0.01). Although SWL and general happiness were significantly related to depression levels (*R*= −0.46 and *R*= −0.48, respectively), no significant relation was observed between these outcomes and GIIS scores. Surprisingly, no relation between changes in reported bodily pain and any other variable (depression, SWL, happiness, or GIIS) was observed. The GDS score at baseline could explain these results. When the GDS score is higher than 5, it is closely related to a diagnosis of depression. Given that the GDS score at baseline was 7.6 in GLG, the relevance of depression symptoms could be very high. The GDS score was reduced to 5.36, which is near the cut-score for depression diagnosis (i.e., 5). Therefore, the reduction of depression levels could be the reason for the difference in the GIIS scores, rather than anything else (including pain).

Surprisingly, no improvement in any dimension of the SF-12 outcome was found, in contrast to the results reported by Zhao et al. [23] in breast cancer patients undergoing endocrine therapy. The limitations of the study could partly explain the differences. First, although statistically significant differences were not found in any of the outcome measures, treatment effects cannot be discarded due to the small sample size. Second, the administered dose could be insufficient to study the efficacy of the treatment. In fact, the dose of GL was selected based on previous studies; however, there is a lack of studies that evaluate the most appropriate dose of GL in adult women. Third, the 6-week intervention period might be too short to see any effect on certain variables, i.e., depression, pain, and HRQoL. Fourth, we could not control the possible interaction between other treatments and GL. Fifth, we did not consider somatic symptoms that might influence depression levels.

Our results confirm the safety of GL. The minor side effects we found were in accordance with those encountered in previous studies [25].

## 5. Conclusions

Our results did not show GL to have any statistically significant effect on happiness, depression, SWL, HRQoL, and the perception of change in women with FMS. However, considering both the improvements found in GLG with respect to the baseline and the study limitations, it would be necessary to carry out more accurate studies to verify the potential of GL in increasing happiness and SWL scores and reducing depression levels.

## Figures and Tables

**Figure 1 healthcare-08-00520-f001:**
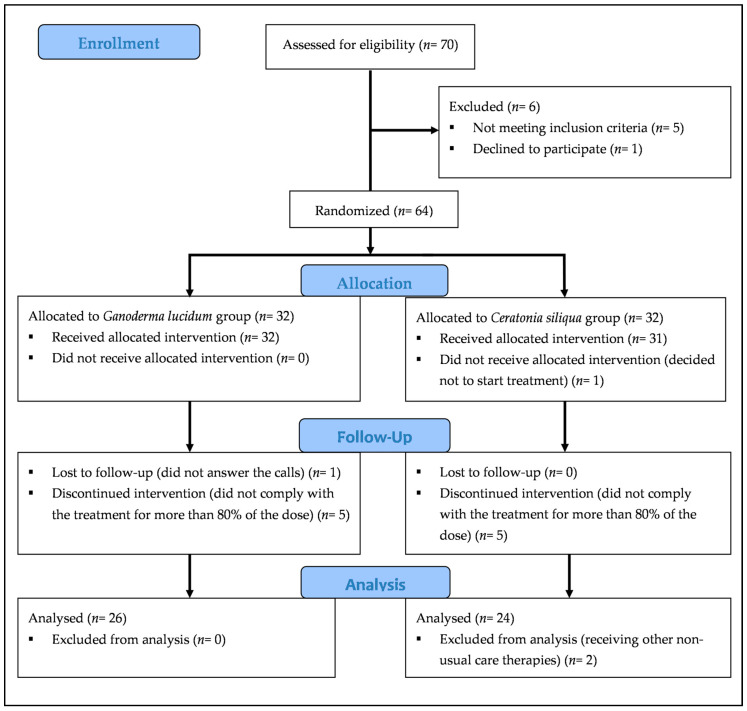
Flow diagram of participants.

**Table 1 healthcare-08-00520-t001:** Characteristics of women with fibromyalgia from the two groups at baseline.

Characteristics	GLG (*n* = 26) ^a^	PG (*n* = 24) ^a^	*p*
Age (years)	56.19 (7.97)	53.74 (11.50)	0.382 *
Date when fibromyalgia symptoms started	1994 (11.86)	1992 (12.56)	0.601 *
Date of diagnosis	2006 (6.56)	2003 (7.02)	0.935 *
Height (cm)	157.08 (4.55)	156.29 (6.08)	0.541 *
Weight (kg)	64.26 (9.67)	61.30 (13.24)	0.411 *
Muscle mass (%)	61.97 (7.17)	64.82 (8.58)	0.245 *
Fat mass (%)	34.81 (7.51)	32.24 (7.75)	0.285 *
BMI (kg/m^2^)	26.05 (3.75)	25.06 (4.75)	0.522 *
Physical Activity (hours per week)	2.96 (1.88)	2.32 (1.43)	0.203 *
Income (euros). GLG: *n* = 23; PG: *n* = 19	1624 (895)	1573 (853)	0.849 *
Type of treatment *n* (%) ^b^			
Number of participants with nonpharmacological treatment	20 (76.9)	18 (75)	0.874 **
Number of participants without nonpharmacological treatment	6 (23.1)	6 (25)	
Number of participants with pharmacological treatment	13 (50)	12 (50)	1.000 **
Number of participants without pharmacological treatment	13 (50)	12 (50)	
Educational Qualifications (%) ^b^			
No education (able to read and write)	4 (15.4)	2 (8.3)	0.048 **
Elementary school	13 (50)	7 (29.2)	
Secondary school	4 (15.4)	13 (54.2)	
University diploma	3 (11.5)	1 (4.2)	
University degree	2 (7.7)	0 (0)	
PhD	0 (0)	1 (4.2)	
Occupational status *n* (%) ^b^			
self-employed	2 (7.7)	0 (0)	0.167 **
paid employment	3 (11.5)	7 (29.2)	
Civil servant	6 (23.1)	2 (8.3)	
Unemployed	6 (23.1)	2 (8.3)	
Retired	3 (11.5)	5 (20.8)	
Housewife	6 (23.1)	7 (29.2)	
Student	0 (0)	1 (4.2)	

^a^ Values expressed as means (SD). ^b^ Values expressed as *n* (%). GLG: *Ganoderma lucidum* group; PG: placebo group; BMI: body mass index; * *p*: value of the Student’s t-test. ** *p*: value of the chi-squared test.

**Table 2 healthcare-08-00520-t002:** Effects of 6-week treatment with *Ganoderma lucidum* or placebo on happiness, satisfaction with life, depression, and GIIS in women with fibromyalgia ^a^.

Outcome Measurements	Baseline (Mean ± SD)	After 6 Weeks’ Treatment (Mean ± SD)	*p* *	Treatment Effect Mean (95% CI)	*p* **
Analysis of the participants who completed the study					
SHS					
GLG (*n* = 26)	3.83 ± 1.57	4.67 ± 1.44	0.009	0.66 (from 0.01 to 1.32)	0.048
PG (*n* = 23)	4.55 ± 1.10	4.74 ± 0.93	0.428		
SWLS					
GLG (*n* = 26)	16.58 ± 7.28	19.27 ± 7.17	0.003	2.69 (from −0.46 to 5.85)	0.092
PG (*n* = 23)	19.13 ± 7.34	19.13 ± 7.31	0.326		
GDS					
GLG (*n* = 25)	7.60 ± 3.39	5.36 ± 3.94	0.001	−1.51 (from −3.51 to 0.48)	0.134
PG (*n* = 22)	6.55 ± 3.12	5.81 ± 3.74	0.379		
GIIS					
GLG (*n* = 26)	NA	2.54 ± 1.45	NA	NA	0.037
PG (*n* = 24)	NA	3.46 ± 1.59			
Intent-to-treat analysis (*n* = 64; GLG = 32; PG = 32)					
GHS					
GHS	4.00 ± 1.64	4.69 ± 1.45	0.007	0.52 (from −0.048 to 1.09)	0.072
GLG	4.43 ± 1.25	4.59 ± 1.14	0.305		
SLS					
SLS	17.22 ± 7.17	19.60 ± 7.35	0.003	3.35 (from 0.62 to 6.09)	0.017
GLG	19.00 ± 6.80	18.03 ± 6.90	0.326		
GDS					
GDS	7.25 ± 3.45	5.60 ± 4.14	0.007	−0.93 (from −3.67 to 0.23)	0.082
GLG	6.31 ± 3.01	6.38 ± 4.14	0.927		

^a^ Values are expressed in points. * *p*: t-test values for the intragroup analysis. ** *p*: values for the analysis of variance for repeated measures to compare the difference between groups after 6 weeks’ treatment (after applying the Bonferroni correction, *p* < 0.004). GDS: Geriatric Depression Scale; SHS: Subjective Happiness Scale; SWLS: Satisfaction with Life Scale; GIIS: Global Impression of Improvement Scale; GLG: *Ganoderma lucidum* group; PG: placebo group; NA: not applicable.

**Table 3 healthcare-08-00520-t003:** Effects of 6-week treatment with *Ganoderma lucidum* or placebo on HRQoL in women with fibromyalgia. Analysis of the participants who completed the study ^a^.

Outcome measurements	Baseline (Mean ± SD)	After 6 Weeks’ Treatment (Mean ± SD)	*p* *	Treatment Effect. Mean (95% CI)	*p* **
SF12v2 (*n* = 50; GLG = 26; PG = 24)					
Physical Function					
GLG	39.42 ± 37.53	47.12 ± 31.09	0.175	5.61 (from −10.48 to 21.70)	0.487
PG	41.67 ± 24.08	43.75 ± 35.55	0.723		
Physical Role					
GLG	52.88 ± 31.29	62.50 ± 28.06	0.106	1.8 (from −14.76 to 18.37)	0.828
PG	46.35 ± 22.57	54.17 ± 27.00	0.200		
Bodily Pain					
GLG	40.38 ± 39.42	56.73 ± 29.63	0.047	10.10 (from −9.64 to 29.84)	0.309
PG	40.63 ± 31.98	46.88 ± 25.87	0.283		
General Health					
GLG	17.50 ± 22.55	30.00 ± 25.22	0.015	4.58 (from −9.13 to 18.29)	0.505
PG	27.08 ± 22.98	35.00 ± 21.37	0.118		
Vitality					
GLG	30.77 ± 31.07	38.46 ± 32.58	0.319	−7.93 (from −26.88 to 11.02)	0.404
PG	26.04 ± 23.86	41.67 ± 20.41	0.008		
Social Functioning					
GLG	51.92 ± 37.36	72.11 ± 38.94	0.030	13.94 (from −10.23 to 38.11)	0.252
PG	56.25 ± 37.77	62.50 ± 36.11	0.450		
Emotional Role					
GLG	61.06 ± 28.79	75.48 ± 26.81	0.033	5.57 (from −12.52 to 23.66)	0.539
PG	59.90 ± 27.82	68.75 ± 26.58	0.174		
Mental Health					
GLG	39.90 ± 25.25	59.61 ± 25.32	0.001	6.17 (from −7.61 to 19.95)	0.373
PG	47.91 ± 19.39	61.46 ± 17.64	0.008		
Standardized Physical Component					
GLG	34.31 ± 9.89	37.31 ± 10.70	0.104	2.24 (from −2.88 to 7.37)	0.383
PG	34.55 ± 7.48	35.31 ± 9.21	0.681		
Standardized Mental Component					
GLG	39.46 ± 10.57	46.33 ± 13.20	0.011	−0.06 (from −6.66 to 6.65)	0.987
PG	39.83 ± 9.65	46.76 ± 9.54	0.003		

^a^ Values are expressed in points. * *p*: values of *t*-test for the intragroup analysis. ** *p*: values for the analysis of variance for repeated measures to compare the difference between groups after 6 weeks′ treatment (after applying the Bonferroni correction, *p* < 0.004). SF12v2: Short-Form Health Survey 12 version 2; GLG: *Ganoderma lucidum* group; PG: placebo group.

**Table 4 healthcare-08-00520-t004:** Symptoms of participants who did not complete the minimum of 80% of treatment.

Group	GLG (*n* = 5)					PG (*n* = 5)				
Participants	P1	P2	P3	P4	P5	P1	P2	P3	P4	P5
Ingested doses before withdrawal	30	30	50	52	58	3	4	26	32	67
**Symptoms**										
Stomach problems (pain, acidity, burning, cramp and not specified discomfort)	x	x	x	x	x	x	x		x	x
Nausea and vomiting		x	x	x			x	x	x	x
Diarrhea	x				x			x		
Dyspepsia				x						
Meteorism				x					x	
Agitation							x			
Dehydration and swelling									x	
Headache							x			
Hypertension							x			
Itching and irritation									x	

GLG: *Ganoderma lucidum* group; PG: placebo group. P: participant; x: presence of a specific discomfort.

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
