# Peer review of "Ganoderma lucidum Effects on Mood and Health-Related Quality of Life in Women with Fibromyalgia"

_healthcare, 2020, doi:10.3390/healthcare8040520_

Round 1
Reviewer 1 Report
This study assessed the effects of Ganoderma lucidum on happiness depression, SWL, HRQoL and perception of change in women with FMS.
The study has several strengths. It targets an important clinical issue of improving mood and quality of life among individuals with fibromyalgia. Additionally, the study utilizes a double blind RCT design, which is a considerable strength.
Along with these strengths, some questions and potential limitations arouse.
- The manuscript would benefit from English language editing/proofing
- The study is heavily limited by a small sample size, with n=24-26 analyzed in each group. This is of particular concern given the authors’ claims that they are testing for “efficacy”, which cannot be established here.
- Related to the point above, what authors frame as the main finding of this paper (group differences in happiness) had a particularly small effect size and was just barely-significant. Of note, this barely significant effect was found without correction for multiple comparisons based on the number of outcomes examined.
- Authors state that their primary hypothesis was differences in happiness, while differences in depression, satisfaction with life, and quality of life were secondary. However, authors provide no clear rationale for making happiness the primary outcome, particularly considering ample research indicating how prevalent and problematic depression symptoms and QoL are in this population. For some readers, this may create the appearance that the primary focus on happiness was in part post-hoc / data driven (considering that this variable was the only one other than impression of improvement with significant group differences) rather than a-priori.
- The choice of the geriatric depression scale as the measure for depressive symptoms may raise some concerns. The age criteria specifies 18+, which questions the fit of a scale specifically developed for geriatric populations. This scale is use far less commonly in FMS research. Furthermore, while the authors justify this by a reduced focus on somatic symptoms, such somatic symptoms are relevant to participants level of depression and are important to capture.
- How was adherence / dose actually consumed monitored and reported?
- 2 participants were excluded for “non-usual care therapies”. Can authors indicate what those therapies were, and why they were deemed unfit for inclusion? Were these participants include in the intent to treat analyses? How did the team monitor changes in usual care?
- There is almost no discussion of the findings in the context of previous research in the discussion section
Minor comments
Page 2 line 47: please spell out HRQoL, as this is the first time this acronym is used.
Reviewer 2 Report
The objective of the paper was to evaluate the effects of Ganoderma lucidum on happiness, depression, satisfaction with life and health-related quality of life in women with fibromyalgia. This study addresses an interesting topic with practical implications. There are, however, some aspects that should be reviewed. In my opinion, would be necessary to resolve some doubts regarding the method and to justify and expand some aspects of the discussion and conclusions.
INTRODUCTION
In the introduction the authors include potential positive effects of Ganoderma lucidum in fibromyalgia patients. However, it would also be necessary to report the side effects or problems that the use of this substance can cause. It would also be convenient to inform, if known, what effects a continued consumption may have.
RECRUITMENT
Could the authors provide more information about how they recruited the participants? Who informed the study participants? The associations? Was there an informational meeting with potential participants?
PARTICIPANTS
I think more information about the participants is needed. It would be necessary to include more information in the general description of the sample. Could the authors provide more information about the sociodemographic variables of the participants? For example: employment, income, with whom they live and educational level.
Likewise, it would also be useful to have some information on the type of treatment that the participants followed regularly (psychological, pharmacological, physical activity).
Minor changes:In participants, the manuscript includes:“4) give written informed consent.”They must put “d) give written informed consent.”
INSTRUMENTS
- It would be convenient include the approximate duration of the evaluations and the maximum and minimum duration.
- Can the authors indicate if the Global Impression of Improvement Scale (GIIS) has been adapted to the Spanish population?
PROCEDURE
- Were the participants given all the doses necessary for the intervention at the same time?
- What information were participants given about the aims of the study when they were encouraged to participate? Was there any relevant information related to the effects of the substance?
- Could there be any risk associated with the consumption of Ganoderma lucidum?
- What kind of doubts did the participants raise when the weekly phone call was made to them? Did the same person always call? How long was this call? Was there a script of questions to ask?
- Was it possible to control whether the participants experienced any stressful events during the weeks of the intervention?
DISCUSSION
- To what do the authors attribute the lack of effects in the SF-12 measures?
- Do the authors have any explanation for the effect of the vitality dimension in the control group?
- Would GL intake be a recommended proposal for all types of patients? In the exclusion criteria, for example, it is included that patients with diabetes were not included. Would there be a specific profile of patients that it could benefit? - On the other hand, it would also be necessary to explain whether prolonged use of GL could produce some undesirable effect or may not be appropriate in certain situations.
- It would be advisable to expand the limitations. For example, the lack of control of some variables (additional treatments, side effects and other unproven effects).
- Further, the authors should deepen further into the implications of the results (advantages over other types of interventions, associated risks, patient profile).
- In addition, the authors should be clear and point out the need to be cautious when considering the implications of the results and conduct more studies to verify the effect of this substance on the physical health and psychological well-being of people with fibromyalgia.
In summary, I think the study is interesting, but it is necessary to provide more information on the issues outlined above.
Round 2
Reviewer 1 Report
I thank the authors for addressing my comments. I feel most issues have been adequately addressed. I would recommend that the paper still go through professional English editing and proofreading, as language issues and typos are still present.
Reviewer 2 Report
I think the authors have included modifications that have improved the article. In my opinion, they have also incorporated information that clarifies some doubts in the methodology and results section. Although the limitations section could be extended, the authors have made an effort to incorporate the reviewers' suggestions.